# CodeBERTScore: Evaluating Code Generation with Pretrained Models of Code

**Shuyan Zhou**[*] **Uri Alon**[*†] **Sumit Agarwal** **Graham Neubig**
Language Technologies Institute, Carnegie Mellon University
{shuyanzh,ualon,sumita,gneubig}@cs.cmu.edu

## Abstract

Since the rise of neural natural-language-to-code models (NL→Code) that can generate long expressions and statements rather than a single next-token, one of the major problems has been reliably evaluating their generated output. In this paper, we propose CodeBERTScore: an evaluation metric for code generation, which builds on BERTScore (Zhang et al., 2020). Instead of encoding only the generated tokens as in BERTScore, CodeBERTScore also encodes the natural language input preceding the generated code, thus modeling the consistency between the generated code and its given natural language context as well. We perform an extensive evaluation of CodeBERTScore across four programming languages. We find that CodeBERTScore achieves a higher correlation with human preference and with functional correctness than all existing metrics. That is, generated code that receives a higher score by Code-BERTScore is more likely to be preferred by humans, as well as to function correctly when executed. We release five language-specific pretrained models to use with our publicly available code. Our language-specific models have been downloaded more than **1,000,000** times from the Huggingface Hub.[1]

## 1 Introduction

Natural-language-to-code generation (NL→Code) has seen sharply growing popularity recently due to the emergence of large language models (LLMs) trained on vast amounts of natural language and code (Chen et al., 2021; Fried et al., 2022; Zhou et al., 2023; Austin et al., 2021; Allal et al., 2023). LLMs have reached such a high NL→Code accuracy that they are now useful for the broad programming audience and actually save

---

[*]Equal contribution
[†] Now at Google DeepMind
[1]The code and data are available at https://github.com/neulab/code-bert-score

developers' time when implemented in tools such as GitHub's Copilot. This sharp rise in LLMs' usability was achieved thanks to their ability to accurately generate *long* completions, which span multiple tokens and even lines, rather than only a single next-token as in early models (Allamanis and Sutton, 2013; Movshovitz-Attias and Cohen, 2013). Nevertheless, evaluating and comparing different models has remained a challenging problem (Xu et al., 2022) that requires an accurate and reliable evaluation metric for the quality of the models' generated outputs, and existing metrics are sub-optimal.

**Existing evaluation approaches** The most common evaluation metrics are token-matching methods such as BLEU (Papineni et al., 2002), adopted from natural language processing. These metrics are based on counting overlapping n-grams in the generated code and the reference code. CrystalBLEU (Eghbali and Pradel, 2022) extends BLEU by ignoring the 500 most occurring n-grams, arguing that they are trivially shared between the prediction and the reference. Nonetheless, both BLEU and CrystalBLEU rely on the lexical *exact match* of tokens, which does not account for diversity in implementation, variable names, and code conventions. Figure 1 shows an example: given the reference code in Figure 1(a), both BLEU and CrystalBLEU prefer (rank higher) *the non-equivalent* code in Figure 1(b) over the functionally equivalent code in Figure 1(c).

CodeBLEU (Ren et al., 2020) attempts to lower the requirement for a lexical exact match, by relying on data-flow and Abstract Syntax Tree (AST) matching as well; nevertheless, valid generations may have different ASTs and data flow from the reference code, which may lead to low CodeBLEU score even when the prediction is correct. Further, *partial* predictions may be useful for a program-

**Reference:**

```
int f(Object target) {
  int i = 0;
  for (Object elem: this.elements) {
    if (elem.equals(target)) {
      return i;
    }
    i++;
  }
  return -1;
}
```

(a) The ground truth reference – find *the index* of `target` in `this.elements`.

**Non-equivalent candidate:**

```
boolean f(Object target) {
  for (Object elem: this.elements) {
    if (elem.equals(target)) {
      return true;
    }
  }

  return false;
}
```

(b) **Preferred by BLEU & CrystalBLEU** – find *whether or not* `target` is in `this.elements`.

**Equivalent candidate:**

```
int f(Object target) {
  for (int i=0; i<this.elements.size(); i++) {
    Object elem = this.elements.get(i);
    if (elem.equals(target)) {
      return i;
    }
  }
  return -1;
}
```

(c) **Preferred by CodeBERTScore** – find *the index* of `target` in `this.elements`.

Figure 1: An intuitive example for the usefulness of CodeBERTScore in measuring generated code: Figure 1(a) shows a reference code snippet in Java. Figure 1(b) and Figure 1(c) show two generated predictions. Among these two candidates and given the reference, both BLEU and CrystalBLEU prefer (score higher) the snippet in Figure 1(b), which *is not* functionally equivalent to the reference, while our proposed CodeBERTScore prefers the code in Figure 1(c), which *is* functionally equivalent to the code in Figure 1(a).

mer, but accepting them may lead to partial code that does not parse, and thus cannot be fully evaluated by CodeBLEU (for example, predicting the first line of a `for` loop, without the loop's body).

*Execution-based evaluation* attempts to address these problems by running tests on the generated code to verify its functional correctness (Chen et al., 2021; Athiwaratkun et al., 2022; Li et al., 2022; Wang et al., 2022; Lai et al., 2022). This provides a direct measure of the functionality of the generated code while being agnostic to diversity in implementation and style. However, execution-based evaluation requires datasets that are provided with hand-written test cases for each example, which is costly and labor-intensive to create; thus, only few such datasets exist. Additionally, executing model-generated code is susceptible to security threats, and thus should be run in an isolated sandbox, which makes it technically cumbersome to work with iteratively.

**Our approach**    In this work, we introduce Code-BERTScore, an evaluation metric for code generation, leveraging self-supervised pretrained models of code such as CodeBERT (Feng et al., 2020), and adopting best practices BERTScore (Zhang

et al., 2020). First, CodeBERTScore encodes the generated code and the reference code independently with pretrained models, *with* the inclusion of natural language instructions or comments. Then, we compute the cosine similarity between the encoded representations of each token in the generated code and each token in the reference code. Finally, the best matching token vector pairs are used to compute precision and recall. Code-BERTScore allows comparing code pairs that are lexically different while taking into account the (1) programmatic- or natural-language-context, if such provided; the (2) contextual information of each token; and (3) implementation diversity. Our approach is illustrated in Figure 2.

**Example**    A concrete example is shown in Figure 1: while BLEU and CrystalBLEU prefer (rank higher) *the non-equivalent* code in Figure 1(b) given the reference code in Figure 1(a), Code-BERTScore prefers the code in Figure 1(c), which *is* functionally equivalent to the reference (Figure 1(a)). We note that in this example, the variable names are identical across all three code snippets. When the variable names of the reference are different than the candidate's, it is *even harder*

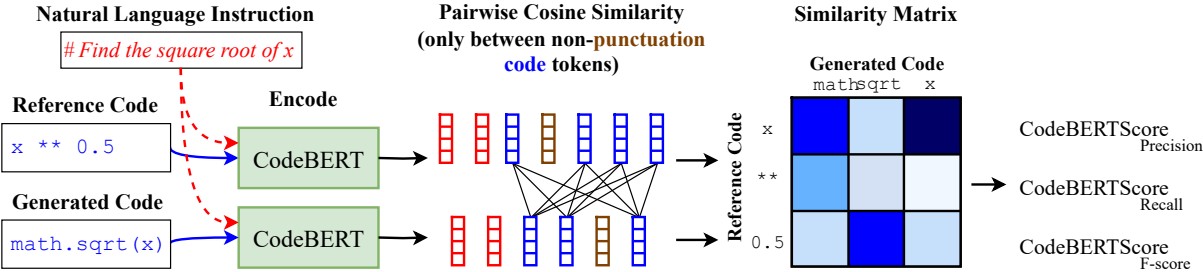

Figure 2: A diagram illustrating CodeBERTScore: We use a language-specific CodeBERT model to encode each of ⟨*natural_language*, *reference_code*⟩ and ⟨*natural_language*, *generated_code*⟩. We then compute the pairwise cosine similarity between every encoded token in the reference and every encoded token in the generated code, ignoring the encoded natural language context tokens and encoded punctuation tokens; finally, we take the $max$ across the rows of the resulting matrix to compute *Precision* and across columns to compute *Recall*.

for token-matching approaches such as BLEU and CrystalBLEU to compare the reference with the candidates, while CodeBERTScore can trivially match variable names according to their semantic similarity and their functional role in the code.

**Contributions**   In summary, our main contributions are: (a) CodeBERTScore: a self-supervised metric for NL→Code evaluation, based on BERTScore, which leverages the benefits of pretrained models, while not requiring labeling or manually annotated data. (b) An extensive empirical evaluation across four programming languages, showing that CodeBERTScore is more correlated with human preference *and* more correlated with execution correctness than all previous approaches including BLEU, CodeBLEU, and CrystalBLEU. (c) We pretrain and release five language-specific CodeBERT models to use with our publicly available code, for Java, Python, C, C++, and JavaScript. As of the time of this submission, our models have been downloaded from the Huggingface Hub more than **1,000,000** times.

## 2   Evaluating Generated Code

### 2.1   Problem Formulation

Given a context $x \in \mathcal{X}$ (*e.g.,* a natural language instruction or comment), a code generation model $\mathcal{M} : \mathcal{X} \rightarrow \mathcal{Y}$ produces a code snippet $\hat{y} \in \mathcal{Y}$ by conditioning on the intent specified by $x$. The quality of the generation is evaluated by comparing $\hat{y} \in \mathcal{Y}$ with the reference implementation $y^* \in \mathcal{Y}$, using a metric function $f : \mathcal{Y} \times \mathcal{Y} \rightarrow \mathbb{R}$, essentially computing $f(\hat{y}, y^*)$.

A larger value of $f(\hat{y}, y^*)$ indicates that the generated code is more accurate with respect to the reference code, and the way $f$ ranks different can-

didates is more important than the absolute value of $f(\hat{y}, y^*)$. That is, ideally, if a prediction $\hat{y}_1$ is more functionally equivalent to $y^*$ and more preferable by human programmers over a prediction $\hat{y}_2$, we wish that a good metric would rank $\hat{y}_1$ higher than $\hat{y}_2$. That is, we seek an $f$ function such that $f(\hat{y}_1, y^*) > f(\hat{y}_2, y^*)$.

### 2.2   Background: BERTScore

BERTScore (Zhang et al., 2020) was proposed as a method for evaluating mainly machine translation outputs. The idea in BERTScore is to encode the candidate sentence (the prediction) and the reference sentence (the ground truth) separately, using a BERT-based model, which encodes each sequence of tokens as a sequence of vectors. Then, BERTScore computes the cosine similarity between every vector from the candidate sequence and every vector from the reference sequences.

Given these similarity scores, BERTScore computes sentence-level precision by taking the maximum similarity score for every candidate vector and averaging, and computes recall by taking the average of the maximum similarity scores for every reference vector. Intuitively, a high BERTScore-recall is obtained, for example, if every vector from the reference sentence has at least one vector from the candidate sentence that is highly cosine-similar to it; a high BERTScore-precision is obtained if every vector from the candidate sentence is highly cosine-similar to at least one vector from the reference sentence. Ultimately, the final score is the $F_1$ score, computed as the harmonic mean of precision and recall.

$$\text{CodeBERTScore}_{\text{P}} \quad = \frac{1}{|\hat{y}[\hat{\boldsymbol{m}}]|} \sum_{\hat{y}_j \in \hat{y}[\hat{\boldsymbol{m}}]} \max_{y_i^* \in y^*[\boldsymbol{m}^*]} sim\left(y_i^*, \hat{y}_j\right) \tag{4}$$

$$\text{CodeBERTScore}_{\text{R}} \quad = \frac{1}{|y^*[\boldsymbol{m}]|} \sum_{y_i^* \in y^*[\boldsymbol{m}^*]} \max_{\hat{y}_j \in \hat{y}[\hat{\boldsymbol{m}}]} sim\left(y_i^*, \hat{y}_j\right) \tag{5}$$

$$\text{CodeBERTScore}_{\text{F}_1} \quad = \frac{2 \cdot \text{CodeBERTScore}_{\text{P}} \cdot \text{CodeBERTScore}_{\text{R}}}{\text{CodeBERTScore}_{\text{P}} + \text{CodeBERTScore}_{\text{R}}} \tag{6}$$

$$\text{CodeBERTScore}_{\text{F}_3} \quad = \frac{10 \cdot \text{CodeBERTScore}_{\text{P}} \cdot \text{CodeBERTScore}_{\text{R}}}{9 \cdot \text{CodeBERTScore}_{\text{P}} + \text{CodeBERTScore}_{\text{R}}} \tag{7}$$

Figure 3: Main equations for CodeBERTScore

## 2.3 CodeBERTScore

Our approach generally follows BERTScore, with the following main differences:

1. We encode the context (the natural language instruction or comment) *along* with each of the generated and reference code snippets, but without using the encoded context in the final similarity computation, essentially computing $f(\hat{y}, y^*, x)$ rather than $f(\hat{y}, y^*)$.

2. Given the precision and recall, instead of computing the $F_1$ score, we also compute $F_3$ to weigh recall higher than precision, following METEOR (Banerjee and Lavie, 2005).

3. As our underlying BERT-like model, we use programming language-specific models that we pretrain and release, rather than models that were intended for natural language only.

We use a BERT-like pretrained model $\mathcal{B}$ to encode the reference and candidate. In our experiments, $\mathcal{B}$ is a CodeBERT model that we further pretrained using the masked language modeling objective (Devlin et al., 2019) on language-specific corpora, but $\mathcal{B}$ can be any transformer-based model which we have access to its internal hidden states.

**Token Representation** We concatenate the context $x$ with each of the reference and the candidate, resulting in $x \cdot y^*$ and $x \cdot \hat{y}$. We use the tokenizer $\mathcal{T}_{\mathcal{B}}$ provided with the model $\mathcal{B}$:

$$\begin{aligned} \mathcal{T}_{\mathcal{B}}\left(x \cdot y^*\right) &= \langle x_1, ..., x_k, y_1^*, ..., y_m^* \rangle \\ \mathcal{T}_{\mathcal{B}}\left(x \cdot \hat{y}\right) &= \langle x_1, ..., x_k, \hat{y}_1, ..., \hat{y}_n \rangle \end{aligned} \tag{1}$$

to get a sequences of tokens. We run a standard "forward pass" with the model $\mathcal{B}$ for each tok-

enized sequence, resulting in sequences of vectors:

$$\begin{aligned} \mathcal{B}\left(\langle x_1, ..., x_k, y_1^*, ..., y_m^* \rangle\right) &= \langle \boldsymbol{x}_1, ..., \boldsymbol{x}_k, \boldsymbol{y}_1^*, ..., \boldsymbol{y}_m^* \rangle \\ \mathcal{B}\left(\langle x_1, ..., x_k, \hat{y}_1, ..., \hat{y}_n \rangle\right) &= \langle \boldsymbol{x}_1, ..., \boldsymbol{x}_k, \hat{\boldsymbol{y}}_1, ..., \hat{\boldsymbol{y}}_n \rangle \end{aligned} \tag{2}$$

Finally, we mask out the encoded context tokens $\boldsymbol{x}_1, ..., \boldsymbol{x}_k$ as well as all non-alphanumeric tokens (parentheses, brackets, dots, commas, whitespaces, etc.) except for arithmetic operators, from each of the encoded reference and encoded candidate. This results in encoded reference tokens $\boldsymbol{y}^* = \langle \boldsymbol{y}_1^*, ..., \boldsymbol{y}_m^* \rangle$, encoded candidate tokens $\hat{\boldsymbol{y}} = \langle \hat{\boldsymbol{y}}_1, ..., \hat{\boldsymbol{y}}_n \rangle$, and their corresponding masks $\boldsymbol{m}^*$ and $\hat{\boldsymbol{m}}$. We denote $\boldsymbol{y}[\boldsymbol{m}]$ as the remaining encoded tokens in $\boldsymbol{y}$ after selecting only alphanumeric token vectors according to the mask $\boldsymbol{m}$.

**Similarity Computation** We compute the cosine similarity between the encoded reference and candidate tokens, following Zhang et al. (2020):

$$sim\left(y_i^*, \hat{y}_j\right) = \frac{\boldsymbol{y}_i^{*\top} \cdot \hat{\boldsymbol{y}}_j}{\|\boldsymbol{y}_i^*\| \cdot \|\hat{\boldsymbol{y}}_j\|} \tag{3}$$

Although this compares the individual tokens $y_i^*$ and $\hat{y}_j$, their vector representations $\boldsymbol{y}_i^*$ and $\hat{\boldsymbol{y}}_j$ contain information about their context, and thus about their semantic role in the code.

**CodeBERTScore** We use the similarity matrix (see Figure 2), formed by the similarity scores between $\boldsymbol{y}^*$ and $\hat{\boldsymbol{y}}$, to compute precision, recall, and $F_1$, by taking the maximum across the rows and columns of the similarity matrix, and then averaging. Following Banerjee and Lavie (2005), we also compute $F_3$ by giving more weight to recall, as shown in Figure 3. Additional details regarding token weighting and scaling are provided in Appendix A.

## 3 Experimental Setup

We evaluate CodeBERTScore across multiple datasets and programming languages. We first show that CodeBERTScore is more correlated with *human preference* than previous metrics, using human-rated solutions for the CoNaLa dataset (Yin et al., 2018a; Evtikhiev et al., 2022). We then show that CodeBERTScore is more correlated with *functional correctness*, using the HumanEval dataset (Chen et al., 2021). We also show that CodeBERTScore achieves a higher newly proposed *distinguishability* than other metrics (Appendix F). Finally, we analyze some of the design decisions and their implications.

### 3.1 Training Language-specific CodeBERT models

**Training** We used CodeBERT (Feng et al., 2020) as our base model ($\mathcal{B}$) and continued its self-supervised pretraining (Gururangan et al., 2020) with the masked language modeling (MLM) objective (Devlin et al., 2019) on Python, Java, C++, C, and JavaScript corpora. We trained a separate model for each programming language, for 1,000,000 steps for each language, using a batch size of 32, an initial learning rate of $5e^{-5}$, decayed linearly to $3e^{-5}$. Our implementation is based on the widely used HuggingFace `Transformers` library (Wolf et al., 2019) and BERTScore[2], and it supports any transformer-based model available on the HuggingFace hub.

**Dataset** We trained each model on the language-specific subset of the CodeParrot (Tunstall et al., 2022) dataset[3], which consists of overall 115M code files from GitHub, further filtered by keeping only files having average line length lower than 100, more than 25% alphanumeric characters, and non-auto-generated files. Even after 1,000,000 training steps, none of the models have completed even a single epoch, meaning that every training example was seen only once at most.

### 3.2 Comparing Different Metrics

We compare CodeBERTScore with existing metrics that are commonly used on code generation evaluation. We use human annotated preference and execution-based results as the ground truth and measure their correlation with these metrics.

**Correlation metrics** We used three major correlation metrics. Following best practices in natural language evaluation, we used Kendall-Tau ($\tau$), Pearson ($r_p$) and Spearman ($r_s$) to measure the correlation between each metric's scores and the references. The detailed equations can be found in Appendix C.

**Human preference experiments** We evaluate different metrics on CoNaLa (Yin et al., 2018b), a natural language to Python code generation benchmark collected from StackOverflow. We use the human annotation of Evtikhiev et al. (2022) to measure the correlation between each metric and human preference. More details are provided in Appendix B.1.

**Functional correctness experiments** We evaluate functional correctness using the HumanEval (Chen et al., 2021) benchmark. Each example in HumanEval contains a natural language goal, hand-written input-output test cases, and a human-written reference solution. While the original HumanEval is in Python, Cassano et al. (2022) translated HumanEval to 18 programming languages, and provided the predictions of the Codex model (Chen et al., 2021) (`code-davinci-002`) and their corresponding functional correctness.[4] We used Java, C++, Python, and JavaScript for these experiments, which are some of the most popular programming languages in open-source projects.[5] More details are provided in Appendix B.2.

**Hyperparameters** We tuned only the following hyperparameters for CodeBERTScore: whether to use $F_1$ or $F_3$, and which layer of the underlying model to extract the encoded tokens from, which we examine in Section 5. We used $F_1$ in the human preference experiments and $F_3$ in the functional correctness experiments. We perform 3-fold cross-validation and report average results across the three folds. As for the layer to extract the token vectors from, we used layer 7 for CoNaLa, and in HumanEval we used layer 7 for Java, 10 for C++, 11 for JavaScript, and 9 for Python.

## 4 Results

**Correlation with human preference** Table 2 shows the correlation between different metrics

---

[2]https://github.com/Tiiiger/bert_score
[3]https://huggingface.co/datasets/codeparrot/github-code-clean
[4]https://huggingface.co/datasets/nuprl/MultiPL-E
[5]https://octoverse.github.com/2022/top-programming-languages

| Metric | Java | | C++ | | Python | | JavaScript | |
|---|---|---|---|---|---|---|---|---|
| | $\tau$ | $r_s$ | $\tau$ | $r_s$ | $\tau$ | $r_s$ | $\tau$ | $r_s$ |
| BLEU | .481 | .361 | .112 | .301 | .393 | .352 | .248 | .343 |
| CodeBLEU | .496 | .324 | .175 | .201 | .366 | .326 | .261 | .299 |
| ROUGE-1 | .516 | .318 | .262 | .260 | .368 | .334 | .279 | .280 |
| ROUGE-2 | .525 | .315 | .270 | .273 | .365 | .322 | .261 | .292 |
| ROUGE-L | .508 | .344 | .258 | .288 | .338 | .350 | .271 | .293 |
| METEOR | **.558** | **.383** | .301 | .321 | .418 | .402 | **.324** | **.415** |
| chrF | .532 | .319 | .319 | .321 | .394 | .379 | .302 | .374 |
| CrystalBLEU | .471 | .273 | .046 | .095 | .391 | .309 | .118 | .059 |
| CodeBERTScore | **.553** | .369 | **.327** | **.393** | **.422** | **.415** | **.319** | .402 |

Table 1: Kendall-Tau ($\tau$) and Spearman ($r_s$) correlations of each metric with the functional correctness on HumanEval in multiple languages. The correlation coefficients are reported as the average across three runs. Standard deviation is provided in Table 3.

| Metric | $\tau$ | $r_p$ | $r_s$ |
|---|---|---|---|
| BLEU | .374 | .604 | .543 |
| CodeBLEU | .350 | .539 | .495 |
| ROUGE-1 | .397 | .604 | .570 |
| ROUGE-2 | .429 | .629 | .588 |
| ROUGE-L | .420 | .619 | .574 |
| METEOR | .366 | .581 | .540 |
| chrF | .470 | .635 | .623 |
| CrystalBLEU | .411 | .598 | .576 |
| CodeBertScore | **.517** | **.674** | **.662** |

Table 2: The Kendall-Tau ($\tau$), Pearson ($r_p$) and Spearman ($r_s$) correlation with human preference. The best performance is **bold**. The correlation coefficients are reported as the average across three runs. Standard deviations are provided in Table 4.

and human preference. CodeBERTScore achieves the highest correlation with human preference, across all correlation metrics. While Evtikhiev et al. (2022) suggested that chrF and ROUGE-L are the most suitable metrics for evaluating code generation models in CoNaLa, CodeBERTScore outperforms these metrics by a significant margin. For example, CodeBERTScore achieves Kendall-Tau correlation of 0.517 compared to 0.470 of chrF and 0.420 of ROUGE-L. These results show that generated code that is preferred by CodeBERTScore— also tends to be preferred by human programmers.

**Correlation with functional correctness** Table 1 shows the correlation between different metrics and functional correctness: CodeBERTScore

achieves the highest or comparable Kendall-Tau and Spearman correlation with functional correctness across *all four* languages. METEOR achieves a comparable correlation with CodeBERTScore in Java and JavaScript, and its correlation is surprisingly better than other baseline metrics. However, in C++ and Python, CodeBERTScore is strictly better. Overall on average across languages, CodeBERTScore is more correlated with functional correctness than all baselines.

## 5 Analysis

We conducted a series of additional experiments to understand the importance of different design decisions, and to gain insights on applying CodeBERTScore to new datasets and scenarios.

**Can we use CodeBERTScore in a new language without a language-specific CodeBERT?** In all experiments in Section 4, we used the language-specific model which we continued to pretrain on each language. But what if we wish to use CodeBERTScore in a language in which we don't have a language-specific model? We compare the language-specific models to CodeBERT-base in Figure 4. Generally, CodeBERT-base achieves close performance to a language-specific model. However, in most HumanEval experiments and correlation metrics, using the language-specific model *is* beneficial. These results show that language-specific models are often preferred if such models are available, but the CodeBERT-base can still provide close performance even without language-specific pretraining.

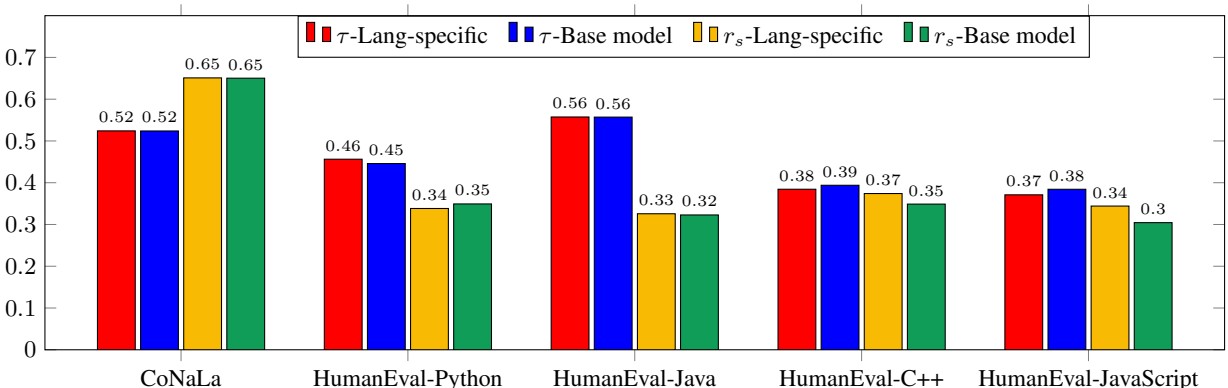

Figure 4: The Kendall-Tau and Spearman on the development set of different datasets with the language-specific pretrained model (Lang-specific) and with the base CodeBERT (Base model).

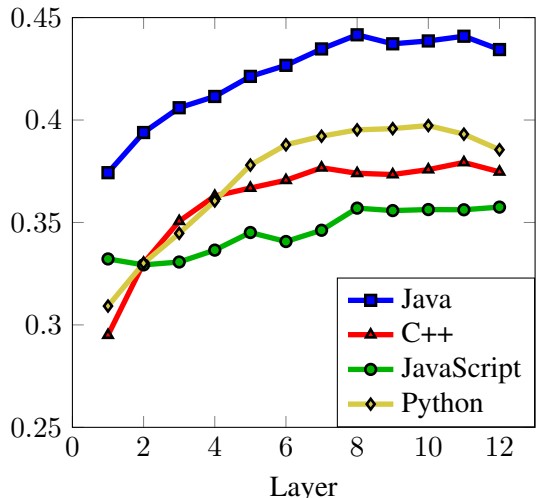

Figure 5: The average of Kendall-Tau and Spearman on the development set of HumanEval when using the embeddings from different layers.

**Which transformer layer should we use?** We further investigate the impact of using hidden states from different layers of the model — the layer which the vectors in Equation (2) come from, in the computation of CodeBERTScore. The results are shown in Figure 5: generally, the deeper the layer – the higher the average correlation between CodeBERTScore and functional correctness, across all programming languages. However in almost all languages, performance reaches its maximum before the last layer, and decreases at the following layers. This suggests that higher layers encode the semantic information of each token more accurately, but the final layers may be more task-specific. These observations are consistent with Tenney et al. (2019), who found that lower layers in BERT tend to process shallow information, while higher layers encode deeper semantic meaning in natural language.

**Does encoding natural language context help?** One major difference between CodeBERTScore and BERTScore is that CodeBERTScore leverages the *context* for the generated code, such as the natural language instruction or intent that was given as input for generation. We find that using context increases the correlation, for example, the Kendall-Tau of CodeBERTScore from 0.50 to 0.52. While this paper mainly focuses on natural language instructions, we believe that Code-BERTScore can thus benefit other programming scenarios as well, for example when generating code given the human-written comments, or generating code given the preceding code context.

**CodeBERTScore allows soft matching of tokens** The heatmaps in Figure 6 show the similarity scores between tokens in CodeBERTScore. For example, both `shutil.rmtree` and `os.rmdir` in Figure 6(a) delete a `folder`; CodeBERTScore aligns each token to a respective token in the other expression, even though the two spans do not share many identical tokens.

In Figure 6(b), both code snippets calculate a square root, where one uses `math.sqrt(x)` and the other uses `x ** 0.5`. An exact surface-form-matching metric such as chrF would assign a low similarity score to this code pair, as they only share the token `x`. However, CodeBERTScore assigns non-zero scores to each token with meaningful alignments, such as matching `[sq, rt]` with `[_0, 5]`, since a square root is the 0.5-th power.

Additionally, we study the robustness of Code-BERTScore to adversarial perturbations. We found that token-based metrics such as chrF are

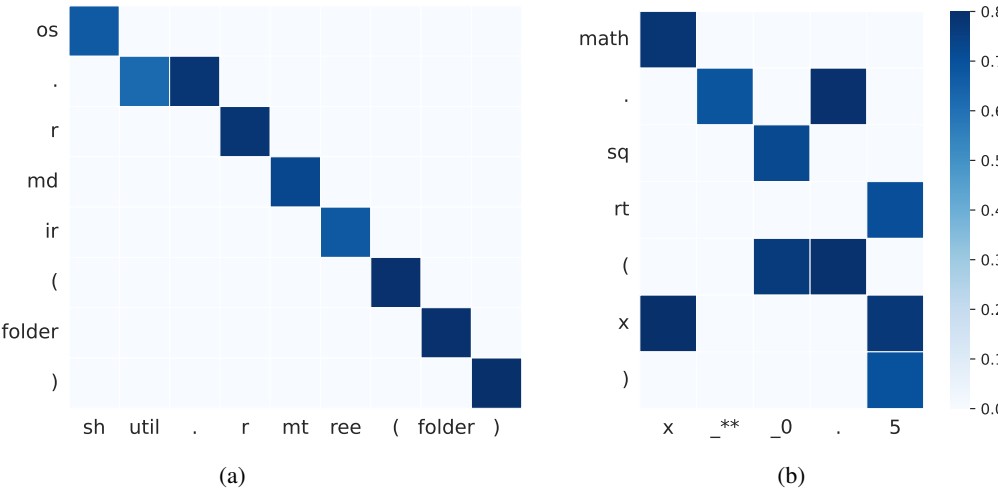

Figure 6: Heatmaps of the similarity scores between two pieces of code that achieve the same goal. Figure 6(a) shows the similarity scores between `os.rmdir(folder)` and `shutil.rmtree(folder)`. Figure 6(b) shows the similarity scores between `math.sqrt(x)` and `x ** 0.5`.

much more prone to matching trivial tokens rather than tokens that preserve the semantic meaning of the code. Examples can be found in Appendix E.

Additional discussion and experiments regarding the distinguishability of CodeBERTScore are provided in Appendix F. Additional general examples are provided in Appendix G.

# 6 Related Work

**Token-based metrics** Metrics such as BLEU (Papineni et al., 2002) evaluate code generation by counting matching n-grams between generated and reference code. CrystalBLEU (Eghbali and Pradel, 2022) refines this approach by disregarding trivially shared n-grams, while ROUGE (Lin, 2004) and METEOR (Banerjee and Lavie, 2005) emphasize recall and balance of precision and recall respectively. However, these metrics, relying on *exact* lexical matches, often fail to capture semantically equivalent but lexically different code snippets. Unlike these, CodeBERTScore captures the wide, two-sided *context* of each token, which n-grams cannot capture.

**Static analysis-based metrics** CodeBLEU (Ren et al., 2020) incorporates data-flow and Abstract Syntax Tree (AST) matching, in addition to token-matching. However, valid code may not always align in ASTs and data-flows. Additionally, partial code, although potentially useful, may not parse, thus cannot be fully evaluated by CodeBLEU. Further, as highlighted by subsequent studies (Wang et al., 2022), CodeBLEU does not correlate well with execution accuracy.

**Execution-based Metrics** To alleviate previous issues, execution-based evaluation counts a generated code snippet as correct if it produces the required outputs when run with given inputs (Chen et al., 2021; Athiwaratkun et al., 2022; Li et al., 2022; Wang et al., 2022; Lai et al., 2022; Huang et al., 2022). However, execution-based evaluation requires datasets that are provided with manually crafted test cases for each example, which is costly and labor-intensive to create; thus, only few such datasets exist. In contrast, CodeBERTScore is completely unsupervised and does not depend on any specific dataset. Further, executing model-generated code is susceptible to security threats, and thus should be run in an isolated sandbox, which makes it technically cumbersome to work with iteratively.

# 7 Conclusion

In this paper, we present CodeBERTScore, a simple evaluation metric for code generation, which builds on BERTScore (Zhang et al., 2020), using pretrained language models of code, and leveraging the natural language context of the generated code. We perform an extensive evaluation across four programming languages which shows that CodeBERTScore is more correlated with human preference than all prior metrics. Further, we show that generated code that receives a higher score by CodeBERTScore is more likely to function correctly when executed. Finally, we

release five programming language-specific pre-trained models to use with our publicly available code. These models were downloaded more than 1,000,000 times from the HuggingFace Hub. Our code and data are available at https://github.com/neulab/code-bert-score.

## Acknowledgement

We thank Misha Evtikhiev, Egor Bogomolov, and Timofey Bryksin for the discussions, and for the data from their paper (Evtikhiev et al., 2022). We thank anonymous reviewers for the valuable feedback. We are grateful to Yiwei Qin for the discussions regarding the T5Score paper (Qin et al., 2022); the idea to use functional correctness as a meta-metric was born thanks to the discussion with her. We are also grateful to Aryaz Eghbali and Michael Pradel for the discussions about CrystalBLEU (Eghbali and Pradel, 2022). This material is partly based on research sponsored in part by the Air Force Research Laboratory under agreement number FA8750-19-2-0200. The U.S. Government is authorized to reproduce and distribute reprints for Governmental purposes notwithstanding any copyright notation thereon. The views and conclusions contained herein are those of the authors and should not be interpreted as necessarily representing the official policies or endorsements, either expressed or implied, of the Air Force Research Laboratory or the U.S. Government. This project was also partially supported by a gift from AWS AI.

## Limitations

CodeBERTScore requires a GPU for computing the metric, while traditional metrics such as BLEU require only a CPU. This adds a hardware requirement to the evaluation of models of code, while most previous approaches are computationally cheaper (*e.g.,* by counting n-grams). However, since training and testing neural models require GPU anyways, we can safely assume that a GPU is available. Further, BERT-base models are encoder-only and non-autoregressive; this means that they require only a *single* "forward pass", compared to encoder-decoder models (*e.g.,* T5) and decoder-only models (*e.g.,* GPT-3) that need to autoregressively generate token after token, using a forward pass for each output token. Thus, the additional time consumption by encoder-only models (*e.g.,* BERT) is negligible, especially when

evaluating encoder-decoder or decoder-only as the NL→Code generator models.

Another point to consider is that CodeBERTScore relies on a strong underlying BERT-based model, while methods such as BLEU do not have many "moving parts" or hyperparameters to tune. However, this is mostly an advantage, since CodeBERTScore can be further improved in the future using stronger base models.

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

## A  Additional Details

**F$_\beta$**  The well-known F$_1$ score is computed as:

$$F_1 = \frac{2}{\frac{1}{\text{recall}} + \frac{1}{\text{precision}}} = \frac{2 \cdot \text{precision} \cdot \text{recall}}{\text{precision} + \text{recall}} \tag{4}$$

A more general F score $F_\beta$ uses a positive factor $\beta$, where recall is considered $\beta$ times as important as precision:

$$F_\beta = \frac{(1 + \beta^2) \cdot \text{precision} \cdot \text{recall}}{\beta^2 \cdot \text{precision} + \text{recall}} \tag{5}$$

As found in METEOR (Banerjee and Lavie, 2005), using F$_\beta$ with $\beta = 3$, thus preferring recall over precision, results in a higher correlation with human preference in machine translation. In our experiments, we found that this applies to NL→Code as well.

**Token Weighting**  Following Zhang et al. (2020), we compute the inverse document frequency (idf), according to a language-specific test set, and weigh each token according to its negative log frequency.

**Scaling**  Following Zhang et al. (2020), the cosine similarity scores of hidden states tend to lie in a limited range. Thus, we can linearly scale the resulting scores, using an empirical base scalar $b$:

$$\widehat{\text{CodeBERTScore}} = \frac{\text{CodeBERTScore} - b}{1 - b} \tag{6}$$

This typically spreads the CodeBERTScore F$_1$ scores to the $[0, 1]$ range, and is merely a cosmetical change: this scaling does not change the way CodeBERTScore *ranks* different prediction, but can be slightly more intuitive and easier to interpret. We computed $b$ empirically by sampling random unrelated code pairs and measuring their average similarity score. For Java, the empirical $b_{\text{Java}}$ was 0.78 and for C++, $b_{\text{C++}}$ it was 0.76.

## B  Evaluation Details

### B.1  Human Preference

For each example, Evtikhiev et al. (2022) asked experienced software developers to grade the generated code snippets from five different models. The grade scales from zero to four, with zero denoting that the generated code is irrelevant and unhelpful, and four meaning that the generated code solves the problem accurately. Overall, there are 2860 annotated code snippets (5 generations × 472 examples) where each snippet is graded by 4.5 annotators.

### B.2  Functional Correctness

We evaluate functional correctness using the HumanEval (Chen et al., 2021) benchmark. Each example in HumanEval contains a natural language goal, hand-written input-output test cases, and a human-written reference solution. On average, each example has 7.7 test cases and there are 164 examples in total. While the original HumanEval is in Python, Cassano et al. (2022) translated HumanEval to 18 programming languages, and provided the predictions of the Codex model (Chen et al., 2021) (code-davinci-002) and their corresponding functional correctness.[6] We used Java, C++, Python, and JavaScript for these experiments, which are some of the most popular programming languages in open-source projects.[7] Notably, Cassano et al. (2022) did not translate the reference solutions to the other languages, so, we collected these from HumanEval-X (Zeng et al., 2022).[8] The reference score of every example is either 1 ("correct", if it passes all test cases) or 0 ("incorrect", otherwise).

## C  Correlation Metrics

**Kendall-Tau ($\tau$)**  $\tau$ measures the *ordinal/rank* association between a metric such as CodeBERTScore and the reference measurement. It is calculated as:

$$\tau = \frac{|\text{concordant}| - |\text{discordant}|}{|\text{concordant}| + |\text{discordant}|}$$

where $|\text{concordant}|$ represents the number of pairs where two measurements agree on their relative rank. That is, if $f(\hat{y_1}, y_1^*) > f(\hat{y_2}, y_2^*)$, the reference measurement also yields $f^*(\hat{y_1}, y_1^*) > f^*(\hat{y_2}, c_2^*)$. Similarly, $|\text{discordant}|$ represents the number of pairs where two measurements yield opposite ranks. Notably, in our experiments, we restrict the comparisons of ranks within the generations of the *same* question.

**Pearson ($r_p$)**  $r_p$ measures the *linear* correlation between a metric and the reference measurement.

---

[6]https://huggingface.co/datasets/nuprl/MultiPL-E
[7]https://octoverse.github.com/2022/top-programming-languages
[8]https://huggingface.co/datasets/THUDM/humaneval-x

It is defined as:

$$r_p = \frac{\sum_{i=1}^{N}(f(\hat{y}_i,y_i^*)-\bar{f})(f^*(\hat{y}_i,y_i^*)-\bar{f^*})}{\sqrt{\sum_{i=1}^{N}(f(\hat{y}_i,y_i^*)-\bar{f})^2 \sum_{i=1}^{N}(f^*(\hat{y}_i,y_i^*)-\bar{f^*})^2}}$$

where $N$ is the number of generations in the dataset, $\bar{f}$ is the mean CodeBERTScore of the dataset, and $\bar{f^*}$ is the mean similarity score calculated by the reference measurement.

**Spearman ($r_s$)** $r_s$ measures the Pearson correlation coefficient between the *ranks* produced by a metric and the reference measurement:

$$r_p = \frac{\text{cov}(R(f(\hat{\mathbf{Y}})), R(f^*(\mathbf{Y}^*)))}{\sigma_{R(f(\hat{\mathbf{Y}}))}\sigma_{R(f^*(\mathbf{Y}^*))}}$$

where $R$ returns the ranks of code snippets in a collection of code snippets $\mathbf{Y}$. $\text{cov}(\cdot,\cdot)$ is the covariance of two variables and $\sigma(\cdot)$ is the standard deviation.

## D  Standard Deviation

Table 3 shows the same results as in Table 1, but with standard deviations. Table 4 shows the results from Table 2, with standard deviations.

## E  Robustness to adversarial perturbations

Ref: `shutil.rmtree(folder)`

| Candidate | CodeBERTScore | chrF |
|---|---|---|
| `os.rmdir(folder)` | 1st | 1st |
| `os.rmdir(f)` | 2nd | 3rd |
| `(folder)` | 3rd | 2nd |

Figure 7: The similarity rankings of three code snippets given the reference code `shutil.rmtree(folder)`. While Code-BERTScore correctly ranks `os.rmdir(f)` over the the non-equivalent `(folder)`, chrF prefers just `(folder)` over `os.rmdir(f)`.

We conducted a qualitative evaluation of Code-BERTScore under various perturbations. An example is shown in Figure 7, which shows the CodeBERTScore and chrF rankings of three code snippets based on the similarity to the reference `shutil.rmtree(folder)`. CodeBERTScore gives a higher ranking to the code snippet that employs the appropriate API (`os.rmdir`) than the trivial `(folder)` that

has the same variable name but without any function call. Contrarily, chrF assigns a higher ranking to `(folder)` which has a longer common sequence of characters, although semantically inequivalent.

## F  Distinguishing Code with Different Semantics

We study how well can CodeBERTScore perform as a generic similarity function that measures the similarity between two arbitrary code snippets $y_i$ and $y_j$.

### F.1  Distinguishability Metric

We evaluate CodeBERTScore using the distinguishability metric $d$ proposed by Eghbali and Pradel (2022) which is calculated as follows:

$$d = \frac{\sum_{y_i,y_j \in \text{Pairs}_{\text{intra}}} f(y_i,y_j)}{\sum_{y_i,y_j \in \text{Pairs}_{\text{inter}}} f(y_i,y_j)} \tag{7}$$

where $\text{Pair}_{\text{intra}}$ defines a set of code pairs from the same semantically equivalent clusters, and $\text{Pair}_{\text{inter}}$ defines a set of code pairs from two clusters of different functionality. Formally,

$$\text{Pair}_{\text{intra}} = \{(y_i,y_j) \mid \exists k \text{ such that } y_i, y_j \in C_k\}$$
$$\text{Pair}_{\text{inter}} = \{(y_i,y_j) \mid \exists k \text{ such that}$$
$$y_i \in C_k, y_j \notin C_k\}$$

where $C_k$ is the $k$-th cluster with semantically equivalent code snippets. Intuitively, a similarity function $f$ that can distinguish between similar and dissimilar code will produce $d$ larger than 1, meaning that a pair of code snippets from the same semantic cluster has a higher similarity score than a pair of snippets from different clusters. Since the number of intra-class and inter-class pairs grows quadratically with the number of code snippets, in our experiments we followed Eghbali and Pradel (2022) to sample $N$ inter- and $N$ intra-class pairs instead.

### F.2  Dataset with Semantically equivalent clusters

We follow Eghbali and Pradel (2022) to evaluate whether CodeBERTScore can distinguish similar and dissimilar code mined from ShareCode[9], an online coding competition platform. Semantically equivalent code snippets are from the same coding problem, and they all pass the unit tests provided

---

[9]https://sharecode.io/

| Metric | Java | | C++ | | Python | | JavaScript | |
|---|---|---|---|---|---|---|---|---|
| | $\tau$ | $r_s$ | $\tau$ | $r_s$ | $\tau$ | $r_s$ | $\tau$ | $r_s$ |
| BLEU | .481(±.030) | .361(±.037) | .112(±.059) | .301(±.054) | .393(±.083) | .352(±.064) | .248(±.075) | .343(±.052) |
| CodeBLEU | .496(±.034) | .324(±.037) | .175(±.021) | .201(±.037) | .366(±.079) | .326(±.075) | .261(±.065) | .299(±.043) |
| ROUGE-1 | .516(±.052) | .318(±.043) | .262(±.073) | .260(±.024) | .368(±.092) | .334(±.054) | .279(±.092) | .280(±.068) |
| ROUGE-2 | .525(±.049) | .315(±.047) | .270(±.073) | .273(±.036) | .365(±.094) | .322(±.077) | .261(±.077) | .292(±.057) |
| ROUGE-L | .508(±.060) | .344(±.038) | .258(±.091) | .288(±.027) | .338(±.103) | .350(±.064) | .271(±.078) | .293(±.046) |
| METEOR | **.558**(±.058) | **.383**(±.027) | .301(±.061) | .321(±.023) | .418(±.090) | .402(±.049) | **.324**(±.075) | **.415**(±.022) |
| chrF | .532(±.067) | .319(±.035) | .319(±.056) | .321(±.020) | .394(±.096) | .379(±.058) | .302(±.073) | .374(±.044) |
| CrystalBLEU | .471(±.024) | .273(±.067) | .046(±.009) | .095(±.064) | .391(±.080) | .309(±.073) | .118(±.057) | .059(±.069) |
| CodeBERTScore | **.553**(±.068) | .369(±.049) | **.327**(±.086) | **.393**(±.048) | **.422**(±.090) | **.415**(±.071) | **.319**(±.054) | .402(±.030) |

Table 3: Kendall-Tau ($\tau$) and Spearman ($r_s$) correlations of each metric with the functional correctness on HumanEval in multiple languages. The correlation coefficients are reported as the average across three runs, along with the standard deviation.

| Metric | $\tau$ | $r_p$ | $r_s$ |
|---|---|---|---|
| BLEU | .374(±.025) | .604(±.016) | .543(±.018) |
| CodeBLEU | .350(±.037) | .539(±.033) | .495(±.037) |
| ROUGE-1 | .397(±.023) | .604(±.016) | .570(±.018) |
| ROUGE-2 | .429(±.025) | .629(±.015) | .588(±.022) |
| ROUGE-L | .420(±.037) | .619(±.014) | .574(±.022) |
| METEOR | .366(±.033) | .581(±.016) | .540(±.022) |
| chrF | .470(±.029) | .635(±.023) | .623(±.018) |
| CrystalBLEU | .411(±.030) | .598(±.019) | .576(±.034) |
| CodeBertScore | **.517**(±.024) | **.674**(±.012) | **.662**(±.012) |

Table 4: The Kendall-Tau ($\tau$), Pearson ($r_p$) and Spearman ($r_s$) correlation with human preference. The best performance is **bold**. The correlation coefficients are reported as the average across three runs. Numbers inside parentheses indicate the standard deviations.

| Metric | Java | C++ |
|---|---|---|
| BLEU | 2.36 | 2.51 |
| CodeBLEU | 1.44 | 1.42 |
| CrystalBLEU | 5.96 | 6.94 |
| CodeBERTScore | **9.56** | **9.13** |

Table 5: Distinguishability with different metrics as the similarity function. CodeBERTScore achieves a higher distinguishability than CrystalBLEU, which proposed this meta-metric, on the same datasets.

by the platform. The dataset consists 6958 code snippets covering 278 problems in Java and C++. We use CodeBERTScore to calculate the similarity score for code pairs that share the same semantic class and code pairs that do not. We then measure the distinguishability of CodeBERTScore according to Equation 7. The results are shown in Table 5.

Table 5 shows that CodeBERTScore achieves a higher *distinguishability* than CrystalBLEU, which proposed this meta-metric, in both Java and C++. CodeBERTScore achieves distinguishability scores of 9.56 in Java while CrystalBLEU achieves 5.96; in C++, CodeBERTScore achieves

9.13 while CrystalBLEU achieves only 6.94. This result confirms that CodeBERTScore assigns higher similarity scores to semantically similar code pairs, compared to randomly paired snippets that belong to different semantic classes.

**Can We Hack the Distinguishability Metric?** Despite the encouraging results in Table 5, we also found that distinguishability can be easily manipulated since it compares *absolute* scores across different metrics. For example, while CrystalBLEU achieves a distinguishability score of 5.96, we can craft a variant of CodeBERTScore that achieves a distinguishability score of 120,000 by simple exponentiation of CodeBERTScore's output score.

To illustrate this, we conducted a distinguishability evaluation with the same configurations as before, but with a variant of CodeBERTScore that we call CodeBERTScore$^k$, and defined as the composition of CodeBERTScore with the $f(x) = x^k$ function, that is: CodeBERTScore$^k(y_1, y_2) = (\text{CodeBERTScore}(y_1, y_2))^k$.

As Figure 8 shows, distinguishability of CodeBERTScore$^k$ increases almost exponentially while increasing $k$, although the base Code-

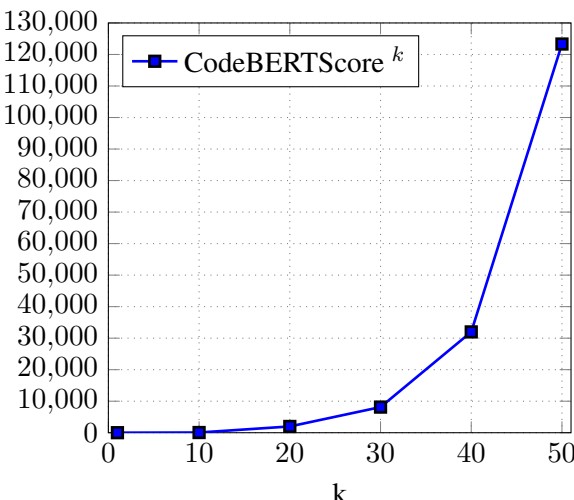

Figure 8: Distinguishability by exponentiating the original CodeBERTScore by $k$.

BERTScore metric has not changed.

We thus argue that distinguishability is not a reliable meta-metric and is no substitute for execution-based- or human-rating. We further suspect that any meta-metric that compares exact, absolute, scores across different metrics is susceptible to such manipulations, and the reliable way to compare metrics is according to the way they *rank* different examples, rather than the exact scores.

The distinguishability results of CodeBERTScore$^k$ with different values of $k$ are shown in Figure 8. As Figure 8 shows, the distinguishability increases almost exponentially with the increasing value of $k$. We thus argue that distinguishability is not a reliable meta-metric and is no substitute for execution-based- or human-rating. We further suspect that any meta-metric that compares exact, absolute, scores across different metrics is susceptible to such manipulations, and the reliable way to compare metrics is according to the way they *rank* different examples, rather than the exact scores.

## G    Additional Examples

In this section, we provide additional examples in which CodeBERTScore prefers the functionally correct prediction, while the best baseline metric in each language ranks higher a functionally incorrect prediction, which is inequivalent to the reference. Figure 9 shows an example in Java, and Figure 10 shows a C++ example.

**Natural Language Question:**

```
/**
Find how many times a given
substring can be found in
the original string.
Count overlaping cases.
>>> howManyTimes("", "a")
0
>>> howManyTimes("aaa", "a")
3
>>> howManyTimes("aaaa", "aa")
3
*/
```

(a) The natural language question.

**Reference:**

```java
public static int howManyTimes(String string,
        String substring) {
  int times = 0;
  for (int i = 0; i < string.length() -
        substring.length() + 1; i++) {
    if (string.substring(i, i + substring.length())
        .equals(substring)) {
      times += 1;
    }
  }
  return times;
}
```

(b) The ground truth reference.

**Non-equivalent candidate:**

```java
public static int howManyTimes(
  String string, String substring) {
  int count = 0;
  int index = 0;
  while ((index = string.indexOf(
        substring, index)) != -1) {
    count++;
    index += substring.length();
  }
  return count;
}
```

(c) **Preferred by METEOR**.

**Equivalent candidate:**

```java
public static int howManyTimes(
    String string, String substring) {
  int counter = 0;
  int index = 0;
  while (true) {
    index = string.indexOf(substring, index);
    if (index == -1)
      break;
    counter += 1;
    index += 1;
  }
  return counter;
}
```

(d) **Preferred by CodeBERTScore**.

Figure 9: An example in HumanEval-Java, in which METEOR assigns a higher score to Figure 9(c) which is *not* functionally equivalent to the reference (Figure 9(b)), while CodeBERTScore prefers the functionally correct candidate Figure 9(d).

**Natural Language Question:**

```
/*
Write a function that takes a vector
of numbers as input and returns the
number of elements in the vector that
are greater than 10 and both first
and last digits of a number are odd
(1, 3, 5, 7, 9). For example:
specialFilter({15, -73, 14, -15})
    => 1
specialFilter({33, -2, -3, 45, 21, 109})
    => 2
*/
```

(a) The natural language question.

**Reference:**

```cpp
long specialFilter(
    std::vector<long> nums) {
  int num=0;
  for (int i=0;i<nums.size();i++)
    if (nums[i]>10) {
        string w=to_string(nums[i]);
        if (w[0]%2==1 and
            w[w.length()-1] % 2 == 1)
          num+=1;
    }
  return num;
}
```

(b) The ground truth reference.

**Non-equivalent candidate:**

```cpp
long specialFilter(
    std::vector<long> nums) {
  // Write your code here
  long sum = 0;
  for(int i=0; i<nums.size(); i++){
    std::string str =
        std::to_string(nums[i]);
    long firstDigit = nums[i] /
        abs(nums[i]);
    long lastDigit = nums[i]%10;
    if(nums[i] > 10 &&
        (firstDigit % 2 == 1 ||
            lastDigit % 2 == 1)){
      sum++;
    }
  }
  return sum;
}
```

(c) **Preferred by chrF**.

**Equivalent candidate:**

```cpp
long specialFilter(
    std::vector<long> nums) {
  long count = 0;
  for(int i=0; i<nums.size(); i++){
    if(nums[i] >= 10) {
      int last = nums[i] % 10;
      int first = nums[i];
      while(first >= 10) {
        first /= 10;
      }
      if (last % 2 != 0 &&
          first % 2 != 0) {
        count++;
      }
    }
  }
  return count;
}
```

(d) **Preferred by CodeBERTScore**.

Figure 10: An example in HumanEval-C++, in which chrF assigns a higher score to Figure 10(c) which is *not* functionally equivalent to the reference (Figure 10(b)), while CodeBERTScore assigns a higher score to the functionally correct candidate Figure 10(d).