# OpenReview forum: "CodeBERTScore: Evaluating Code Generation with Pretrained Models of Code"
_EMNLP/2023/Conference — EMNLP 2023 Main_

### Official Review · Reviewer_Q9Kk · 2023-08-05

**Soundness:** 2

**Ethical Concerns:**

Yes

**Excitement:**

2: Mediocre: This paper makes marginal contributions (vs non-contemporaneous work), so I would rather not see it in the conference.

**Justification For Ethical Concerns:**

As I mentioned above, there I am not sure if the anonymity is not spoiled by the authors in the paper.

**Paper Topic And Main Contributions:**

The paper is devoted to presentation of CodeBERTScore model(s). The authors provide some evaluation of the models with comparison with other existing metrics.


**Reasons To Accept:**

The authors claim that their model is pretty popular on HuggingFace.

**Reasons To Reject:**

I am not sure if it is not violating anonymity to claim that your model has more than 1M downloads on HuggingFace alongside with providing the name.
Another important point is that the authors claim that their score is correlated with human human judgement although in fact it is only the correlation with HumanEval labelling and not independent evaluation.
At last but not least there is no scientific novelty in their work. I could consider this paper as a demo one, but not for the main track.

**Reproducibility:**

4: Could mostly reproduce the results, but there may be some variation because of sample variance or minor variations in their interpretation of the protocol or method.

**Reviewer Confidence:**

4: Quite sure. I tried to check the important points carefully. It's unlikely, though conceivable, that I missed something that should affect my ratings.

---

> ### Author Rebuttal · Authors · 2023-08-28
>
> Thank you for taking the time to review our paper!
> We think that all your questions are addressable within this discussion period. Please see our response below.
>
> > ### The authors claim that their score is correlated with human human judgement although in fact it is only the correlation with HumanEval labelling and not independent evaluation
>
> We did perform human judgment experiments, with the annotations taken from Evtikhiev et al. (2022). The setup is explained in L313-L320; the experimental results are in Table 2; and the discussions are in L350-L364. More details are provided in Appendix B.1.
>
>
>
> > ### What is the novelty of this work?
>
> The main novelty in our work lies in the empirical results demonstrating the utility of (Code)BERTScore (and the new code BERT models that we train) for the evaluation of both human preferences and functional correctness of code. Such a metric is urgently needed in the NLP-SE community, and the community so far did not have any strong metric to rely on which was extensively tested and evaluated. We hope that our work will inspire future work in this direction.
>
> > ### Does mentioning that the model has more than 1M downloads on HuggingFace violate anonymity?
>
> Apologies if mentioning that the model has been downloaded on Hugging Face seemed to break anonymity in some way. This was certainly not our intention, and honestly we don’t think it breaks anonymity any more than providing the model name itself (which is standard in many papers). In either case, it would require the reviewers to **actively search** for the model name, which reviewers are encouraged not to do by the ACL reviewing guidelines: https://acl-org.github.io/policies/reviewers

---

### Official Review · Reviewer_HHa9 · 2023-08-05

**Soundness:** 4

**Excitement:**

4: Strong: This paper deepens the understanding of some phenomenon or lowers the barriers to an existing research direction.

**Paper Topic And Main Contributions:**

This paper extends a well know metric for evaluating text generation, BERTScore, to code generation. Instead of encoding only the generated tokens as in BERTScore, CodeBERTScore also encodes the natural language input preceding the generated code.

CodeBERTScore encodes the generated code and the reference code independently with pretrained models, with the inclusion of natural language instructions or comments. Then, it computes the cosine similarity between the encoded representations of each token in the generated code and each token in the reference code. Finally, the best matching token vector pairs are used to compute precision and recall.

They further pretrain CodeBERT for separate programming languages (Python, Java, C++, C and JS) on CodeParrot and release the models on HuggingFace.

They evaluate across 4 programming languages. To show correlation with human preference (via Kendall-Tau, Pearson and Spearman scores)  they use CoNaLa, to show correlation with functional correctness they use HumanEval.

They find that CodeBERTScore achieves a higher correlation with human preference and with functional correctness (via unit tests) than other standard metrics. Results show significant gains for human preference and gains for functional correctness (in particular for C++ and Python).

**Reasons To Accept:**

1. Metrics for code generation are still poor and the authors improve the SoTA in this space. It is very easy to reuse their work and I believe many researchers and practitioners would benefit from it.

2. The evaluation is thorough and well thought out. They compare to many other metrics and they report sensible estimates for the level of correlation.

3. The paper is well written and easy to follow.

**Reasons To Reject:**

I really enjoyed reading this paper, but the contribution is incremental compared to prior work (they straightforwardly extend BERTScore).

**Reproducibility:**

5: Could easily reproduce the results.

**Reviewer Confidence:**

4: Quite sure. I tried to check the important points carefully. It's unlikely, though conceivable, that I missed something that should affect my ratings.

---

> ### Author Rebuttal · Authors · 2023-08-28
>
> Thank you for taking the time to review our paper and for your kind words!  We were happy to read that you think that “The evaluation is thorough and well thought out” and that you “believe many researchers and practitioners would benefit from it”.
>
> > ### I really enjoyed reading this paper, but the contribution is incremental compared to prior work (they straightforwardly extend BERTScore).
>
> The main novelty in our work lies in the empirical results demonstrating the utility of (Code)BERTScore (and the new code BERT models that we train) for the evaluation of both human preferences and functional correctness of code. Such a metric is urgently needed in the NLP-SE community, and the community so far did not have any strong metric to rely on which was extensively tested and evaluated. We hope that our work will inspire future work in this direction.

---

### Official Review · Reviewer_29mb · 2023-08-10

**Typos Grammar Style And Presentation Improvements:** None.
**Soundness:** 4

**Excitement:**

5: Transformative: This paper is likely to change its subfield or computational linguistics broadly. It should be considered for a best paper award. This paper changes the current understanding of some phenomenon, shows a widely held practice to be erroneous in someway, enables a promising direction of research for a (broad or narrow) topic, or creates an exciting new technique.

**Missing References:**

None.

**Paper Topic And Main Contributions:**

Topic: This paper addresses the problem of evaluating the code generated by neural network, which needs achieve a higher correlation with human preference and with functional correctness than all existing metrics.

Contributions:
A. The introduction of CodeBERTScore. CodeBERTScore builds upon BERTScore, a contextual encoding method, to compute a soft similarity score between the generated code and the reference code. Unlike traditional metrics like BLEU, which rely on exact token matching, CodeBERTScore takes into account the contextual encodings of large pretrained models. Additionally, CodeBERTScore encodes not only the generated tokens but also the programmatic context surrounding the generated code.
B. Large-scale empirical research. The authors extensively evaluate CodeBERTScore across four programming languages. They demonstrate that CodeBERTScore achieves a higher correlation with human preference and functional correctness compared to existing metrics. Moreover, they conducted a series of additional experiments to understand the importance of different design decisions, and to gain insights on applying CodeBERTScore to new datasets and scenarios.

**Questions For The Authors:**

I believe that your paper is well experimented and well written. But I still have a few questions that I need you to clarify.

A. About Human Preference.
In the paper, the authors mention that "use the human annotation released by Evtikhiev et al. (2022) to measure the correlation between each metric and human preference."
I am confused about this, why don't the authors use the same method to evalute on other datasets (not just CoNaLa)?

B. The choice of base models and pre-training methods.
Why do authors focus on CodeBERT and not consider alternatives such as GraphCodeBERT and UniXcoder. Additionally, it is relevant to inquire why pre-training tasks like contrast learning were not taken into account.

C. Robustness.
Did the authors design CodeBERTScore with its robustness in mind?

**Reasons To Accept:**

A. Relevance: Code generation is an important and challenging task in the field of natural language processing and software enginering. This paper addresses the crucial problem of evaluating the output of code generation models, which is of great interest to the NLP and SE.

B. Evaluation: The paper provides extensive evaluation of CodeBERTScore across multiple programming languages, demonstrating its higher correlation with human preference and functional correctness compared to existing metrics. This evaluation adds to the credibility and significance of the proposed metric.

C. Practicality and Open Access: The authors make their code and pre-trained models publicly available, facilitating reproducibility and remedying the gap in the code generation field where semantic-based evaluation metrics are missing. This practical contribution provides an invaluable resource for researchers and developers in NLP and SE.

**Reasons To Reject:**

A. Novelty: While CodeBERTScore builds upon the BERTScore approach and applies it to the code generation domain, it may not introduce significant novelty compared to applying BERTScore to other languages (such as Japanese or Russian languages). However, it is still valuable to have a dedicated evaluation metric for code generation, as it addresses the unique challenges and requirements of that task.

B. Customization: CodeBERTScore is not customised for the features of the code (e.g., syntax information, data flow information, or even LLVM-IR information obtained through the compiler).

C. Robustness: We are aware that existing neural networks and pre-trained models may encounter challenges with robustness. While I found the performance of CodeBERTScore to be impressive when I downloaded and used it, I did come across potential issues.
Firstly, CodeBERTScore appears to have difficulty accurately recognizing different implementations of the same functional code, such as distinguishing between looping statements with "for" and "while" constructs.
Secondly, CodeBERTScore seems to struggle when it comes to code that contains similar tokens but different functions. For example, it may not effectively differentiate between expressions like 'a += b' and 'a -= b'.
These observations imply that CodeBERTScore may lack robustness in accurately understanding the semantics of code.

**Reproducibility:**

5: Could easily reproduce the results.

**Reviewer Confidence:**

5: Positive that my evaluation is correct. I read the paper very carefully and I am very familiar with related work.

---

> ### Author Rebuttal · Authors · 2023-08-28
>
> Thank you for taking the time to review our paper and for your kind words!  We were happy to read that you think that this paper is “of great interest to NLP and SE” and that “The paper provides extensive evaluation”.
>
> We think that all your questions are addressable within this discussion period. Please see our response below.
>
> > ### While I found the performance of CodeBERTScore to be impressive when I downloaded and used it ... CodeBERTScore may lack robustness in accurately understanding the semantics of code ...
>
> This is a great point. We agree that distinguishing code snippets with nuance differences is challenging for all metrics, including CodeBERTScore. Quantitatively and qualitatively (Figure 6 and Appendix G), we observe that CodeBERTScore is more robust and accurate than all previous methods, including approaches that explicitly take data dependency into account (such as CodeBLEU).
>
> We agree that further improving the robustness of neural-based metrics is an important direction for future research. To this point, we are planning to include a discussion and analysis of the current limitations in the revised version, specifically adding typical examples where CodeBERTScore still fails at providing good assessments. Thank you for raising this point, we believe this additional analysis will make the paper stronger!
>
>
>
> > ### CodeBERTScore is not customised for the features of the code (e.g., syntax information, data flow information, or even LLVM-IR information obtained through the compiler).
>
> Although CodeBERTScore doesn’t explicitly encode syntax or data-flow information, we compare it with metrics that do so, such as CodeBLEU which uses data-flow and Abstract Syntax Trees. However, as shown in Table 1 and Table 3, CodeBLEU does not exhibit a superior correlation with human preference and functional correctness, and is sometimes even lower than BLEU.
>
> We believe that similarly to the process that NLP research has gone through, from using syntax, dependency, etc. to pure neural modeling, it might be the case for code as well: although syntax and data flow information are provided “for free”, strong neural models can just work better empirically. This is also evident by the rise of generative models of code such as Codex and  CodeLlama, which perform much better than prior models that explicitly generated ASTs and graphs.
>
> > ###  it may not introduce significant novelty compared to applying BERTScore to other languages ... However, it is still valuable to have a dedicated evaluation metric for code generation, as it addresses the unique challenges and requirements of that task.
>
> The main novelty in our work lies in the empirical results demonstrating the utility of (Code)BERTScore (and the new code BERT models that we train) for the evaluation of both human preferences and functional correctness of code. Such a metric is urgently needed in the NLP-SE community, and the community so far did not have any strong metric to rely on which was extensively tested and evaluated. We hope that our work will inspire future work in this direction.
>
> > ### Q1: the authors mention that they "use the human annotation released by Evtikhiev et al. (2022) to measure the correlation between each metric and human preference." I am confused about this, why don't the authors use the same method to evaluate on other datasets?
>
> The annotations released by Evtikhiev contain only annotations for the Conala dataset. We thus used functional correctness to evaluate the different metrics on the other dataset (HumanEval).
>
>
> > ###  Q2: Why do authors focus on CodeBERT and not consider alternatives such as GraphCodeBERT and UniXcoder? Additionally, it is relevant to inquire why pre-training tasks like contrast learning were not taken into account
> > ### Q3: Did the authors design CodeBERTScore with its robustness in mind?
>
> Since such an approach with its extensive evaluation was so missing in the current literature and the community, we wanted to show that **even the simplest base model helps**.
> Our approach is certainly not coupled with the specific “CodeBERT” model. The underlying base model can be replaced easily with any model that is available on the Huggingface hub.
> We believe that our paper presents the first extensive evaluation in this direction, and that stronger base models and better training, such as contrastive learning, can certainly further improve the results.

---

### Meta-Review · Area_Chair_jnRC · 2023-09-01

**Recommendation:** Accept (Poster)
**Confidence:** 5

**Metareview:**

Model generated code evaluation is a really hard problem and this work is a very interesting and useful contribution to that issue. While the authors go to good lengths to justify the value of the model, it is still perhaps a matter of judgement whether the score coming from CodeBERTScore is more meaningful than other metrics. It's my opinion that code execution in environments with tests is the gold standard for code evaluation, but this is a very high burden for many important questions.

---

### Decision · Program_Chairs · 2023-10-07

**Decision:**

Accept-Main

**Comment:**

Model generated code evaluation is a really hard problem and this work is a very interesting and useful contribution to that issue. While the authors go to good lengths to justify the value of the model, it is still perhaps a matter of judgement whether the score coming from CodeBERTScore is more meaningful than other metrics. It's my opinion that code execution in environments with tests is the gold standard for code evaluation, but this is a very high burden for many important questions.